# Emerging Diseases in Spain Strawberry Crops: Neopestalotiopsis Leaf and Crown Rot and Fusarium Wilt

**DOI:** 10.3390/plants13233441

**Published:** 2024-12-08

**Authors:** Manuel Avilés, Ana M. Pastrana, Celia Borrero

**Affiliations:** 1Departamento de Agronomía, ETSIA-Universidad de Sevilla, Ctra. Utrera Km 1, C.P, 41013 Seville, Spain; apastrana@us.es (A.M.P.); cborrero@us.es (C.B.); 2Center Imperial County, University of California Cooperative Extension, 1050 Holton Rd., Holtville, CA 92250, USA

**Keywords:** strawberry, *Fusarium oxysporum* f. sp. *fragariae*, pathotypes, races, inoculum sources, cultivar susceptibility, chemical control, biological control

## Abstract

In recent years, strawberry cultivation in Spain has been increasingly affected by new and re-emerging fungal diseases. The most significant emerging diseases in Spain include those caused by *Neopestalotiopsis* spp. Maharachch., K.D.Hyde & Crous and *Fusarium oxysporum* f. sp. *fragariae* Winks & Y.N. Williams. These pathogens are difficult to control due to their pathogenic variability (presence of pathotypes and/or races), the lack of knowledge about the susceptibility of the different cultivars, the limited availability of effective fumigants, and the absence of sufficient information about their sources of inoculum. Both pathogens can cause root and crown rot, leading to plant collapse and significant losses for strawberry producers. Several factors have contributed to the rise of these diseases in Spain: (i) the gradual ban on key soil fumigants has left the crop vulnerable; (ii) there has been a notable diversification in the origin of mother plants used in cultivation, with plants now sourced from various countries, increasing the risk of long-distance pathogen spread; (iii) the introduction of numerous new strawberry varieties, which exposes more genotypes to pathogenic infections; and (iv) changes in planting times, leading to younger and more vulnerable plants being exposed to heat stress, as well as an increase in disease susceptibility. *Neopestalotiopsis* spp. and *Fusarium oxysporum* f. sp. *fragariae* have also become major threats to strawberry crops worldwide, spreading through nursery plants and the movement of plant material. The latest research findings in Spain on both pathogens are highlighted in this manuscript.

## 1. Introduction

Spain is the leading producer of strawberry plants and fruit (*Fragaria* × *ananassa* Duch.) in Europe [1], being a crop of great economic and social importance. Strawberry cultivation is divided both professionally and geographically. Thus, the production of plants is carried out in high-altitude nurseries in Castilla y León, while the cultivation for obtaining fruit is carried out in the fruit production fields in Huelva (Figure 1). In the last season, around 1500 hectares of strawberry nurseries were established in Castilla y León, with a production of around 1000 million plants [2]. At the nursery, the crop is grown outdoors and extensively. After the uprooting or harvesting of the daughter plants, during September and October, they are sent to the fruit production areas where they are transplanted. In Spain, the cultivation of strawberry fruit is concentrated in the coastal area of Huelva and is characterized by being part of an intensive and technified horticulture, with a marked export projection. In the 2022/23 season, Huelva produced 97% of the Spanish strawberry market and almost 29% of the EU-27 strawberry market, making Spain the leading strawberry producer in the EU [1]. The value of the production of this fruit in Andalusia in 2022 reached EUR 407.83 million. The average annual production is around 325 t of fruit, of which 80–85% is destined for the fresh market in the European Union, the main receiving countries being Germany, France, the United Kingdom, and Italy [3].

In countries where strawberries are grown, there has been an observed emergence of new fungal diseases and a resurgence of previously nearly eradicated ones in recent years. Soilborne fungi, in particular, are now recognized among the most limiting plant pathogens for this crop. In Spain, the most significant fungal diseases affecting the crown and roots, which can result in plant death, include anthracnose caused by *Colletotrichum acutatum*, Phytophthora crown and root rot caused by *Phytophthora cactorum*, and charcoal rot caused by *Macrophomina phaseolina*. Additionally, leaf and crown rot caused by *Neopestalotiopsis* spp. and Fusarium wilt caused by *Fusarium oxysporum* f. sp. *fragariae* have emerged as significant diseases (Figure 1). There are no data on the potential economic impact of these emerging diseases in Spain, and even less data separated by disease. However, growers and technicians often estimate a reduction in crop yield of 10-25% due to the incidence of the various diseases that cause root and crown rot. This could result in economic losses of between EUR 40 and 100 million per year. Both emerging diseases present challenges for diagnosis and control due to the presence of pathotypes, the lack of information on inoculum sources, varying susceptibility among cultivated varieties, and the absence of effective authorized fumigants and fungicides for their treatment.

From an agronomic perspective, we highlight four relatively recent changes with potential implications for the emergence of these diseases in Spain and possibly in other producing regions:(i)The gradual ban on the most commonly used soil fumigants, such as methyl bromide (MB), 1,3-dichloropropene, and chloropicrin. This crop was highly dependent on MB for soil disinfestation. Following the Montreal Protocol and European Community Regulation (EC) No. 2037/2000 of June 29 on substances that deplete the ozone layer, the ban on MB for strawberry production became irreversible in EU countries from 2007 (EC Commission Decision, 2006). This ban was later extended to other fumigants like 1,3-dichloropropene and chloropicrin. Currently, in Spain, only the commercialization and use of metam sodium formulations are exceptionally authorized, limited to application once every three seasons [4].(ii)Increased diversification in the origins of mother plants (F0) and transplant plants (F3). Strawberry plant production, which eventually leads to transplantation, starts with F0 plants. These plants, previously sourced mainly from the USA and one or two breeding programs, now come from various origins and multiple breeding programs, which are primarily national but also originate from other countries. From these F0 plants, F1 plants are derived, which then produce daughter plants (F2) grown in high-altitude nurseries. These F2 plants are extensively cultivated to eventually obtain transplant plants (F3). While transplant plant production is mainly national, there is also importation from countries like Poland. The inherent national and international movement of plant material for production facilitates long-distance distribution of plant pathogens.(iii)A significant varietal turnover, with the continuous increase in the number of varieties cultivated in Huelva (Spain). The number of varieties with more than 1% of the planted area increased from 11 in 2019/20 to 22 in 2022/23 [5]. This results in a greater number of genotypes exposed to a range of pathogens and pathotypes.(iv)The trend towards earlier transplanting dates, resulting in smaller and more tender plants that are less resistant to transplant stress. Moreover, earlier planting increases the likelihood of high-temperature events during establishment. All these factors may contribute to greater susceptibility to various diseases.

## 2. Leaf Spot and Root and Crown Rot in Strawberry Caused by *Neopestalotiopsis* spp.

Until recently, *Neopestalotiopsis* spp. was considered a secondary, weak, and opportunistic pathogen, almost always isolated from crowns and roots along with more aggressive fungi such as *Colletotrichum* spp., *Phytophthora cactorum*, and *Macrophomina phaseolina*. However, since the first reported case of crown and root rot in Huelva province [6], the incidence of *Neopestalotiopsis* spp. in strawberry plants has been increasing, currently representing one of the main problems for Spanish producers. It has also been found to cause leaf spots both in nursery plants and in fruit production fields in Huelva. In our diagnostic laboratory, *Neopestalotiopsis* spp. are among the pathogens most frequently isolated from strawberry plant samples we have received since 2020 (Figure 2a,b).

Similarly, in recent years, the pathogen has been described as causing crown and root rot in various production areas such as Argentina [7], Bangladesh [8], Belgium [9], California [10], China [11], Ecuador [12], Egypt [13], Florida [14], Italy [15], Mexico [16], and Uruguay [17]. More recently, it has been identified as a cause of leaf spot and/or fruit rot in several strawberry-producing regions such as Bangladesh [18], China [19], Florida [20], India [21], Indiana [22], Mexico [16,23], Ohio [24], Taiwan [25], and Turkey [26].

### 2.1. Syndromes

Thus, two types of syndromes are distinguished: (i) Crown and root rot with darkening of the roots (Figure 3a) and orange-brown necrotic lesions visible when cutting the crowns (Figure 3b), contributing to stunting and poor establishment after transplanting (Figure 4). In the aerial parts, wilting and necrosis of older leaves are observed, often resulting in plant collapse and death [16]. (ii) Leaf spot and/or fruit rot appearing as lesions of varying size, dry, bronze-colored, with darker, irregular borders, on leaves and stems (Figure 5).

These lesions increase and eventually become covered with numerous black conidiomata (Figure 6a). Subsequently, these diseased plants may also collapse with crown and root rot (Figure 6b). 

Symptoms on strawberry fruit include 2 to 4 mm diameter lesions that are slightly sunken, dry, light tan in color and irregularly shaped at first. The lesions grow and finally become covered with numerous conidiomata [20].

### 2.2. Pathogenic Diversity

The most recent and aggressive disease outbreaks appear to belong either to *Neopestalotiopsis rosae* (in Mexico) or to a new, significantly more aggressive species closely related to *N. rosae* but yet to be defined (in Florida, USA) [16,20]. Additionally, other *Neopestalotiopsis* species such as *N. clavispora*, *N. mesopotamica*, *N. iraniensis*, and *N. fragariae* have been reported in other strawberry production areas [27]. In preliminary varietal resistance trials with different *Neopestalotiopsis* spp. isolates, we have observed some variability in the susceptibility of the studied cultivars, with different responses in the same cultivar depending on the isolate. This suggests variability in cultivar susceptibility and pathogenic variability. This complicates diagnosis as pathotypes must be differentiated to refine the measures to be adopted by farmers, given the differences in aggressiveness.

### 2.3. Sources of Inoculum

It has been suggested that the primary source of inoculum for diseases associated with *Neopestalotiopsis* spp. is transplant plants from nurseries [16,20,28]. In addition, there is also evidence that *Neopestalotiopsis* spp. can survive in the soil during the summer until the next growing season [29]. Another possible source of primary inoculum could be alternative hosts. Our team previously established that *Neopestalotiopsis clavispora* is pathogenic in blueberries in Huelva [30], and raspberries and blackberries have also been cited as hosts of *Neopestalotiopsis* spp. [31]. These crops often border strawberry fields on many farms in Huelva. Indeed, *Neopestalotiopsis* spp. species have a broad host range. For instance, reservoirs of the pathogen have been found in wild rhododendrons near strawberry nurseries in North Carolina (USA), suggesting that they could serve as a source of genetic variability for the pathogen by completing the sexual cycle on these hosts [20]. Among the species frequently found on the edges and nearby high-altitude nurseries in Castilla y León and cited as hosts of various species of the Pestalotioid fungal taxon (*Seiridium*, *Neopestalotiopsis*, *Pestalotiopsis*, and *Pseudopestalotiopsis*) are pines (*Pinus pinaster* and *Pinus pinea*) and holm oaks (*Quercus ilex*) [32,33,34,35]. These potential inoculum sources have not been evaluated or quantified in our country.

Another relevant consideration is understanding how the pathogen spreads among plants both in nurseries and in the fields of Huelva. Thus, the activities of uprooting, cutting, and handling plants in nurseries (Figure 7) can facilitate the spread of conidia and infection through the wounds produced. In fruit production fields, water splashes and wind, especially on the days following planting when sprinkler irrigation is used, can coincide with warm temperatures suitable for the plant pathogen. Additionally, workers during harvesting and cleaning operations can also contribute to pathogen dispersal. Thus, in our laboratory, 742 qPCR analyses for *Neopestalotiopsis* spp. were performed over the last two seasons on both symptomatic and asymptomatic plant samples, in which 181 positive cases were detected from various Spanish nurseries and fruit production fields, representing approximately 24% of the samples analyzed. This may give an idea of the extent of the outbreak (Figure 1).

One option to mitigate disease spread from nurseries is to apply fungicides preventively to plants prior to transplanting. The most effective fungicides evaluated in other production areas include fludioxonil, fluzinam, prochloraz, captan, thiram, and chlorothalonil, and the mixtures pydiflumetofen + fludioxonil and cyprodinil + fludioxonil, with preventive applications being more effective than curative ones [27,36]. However, in Spain, there are no fungicides authorized for this use, and most of those mentioned are excluded from the annex of Regulation (EC) No. 1107/2009. Among the biofungicides with positive control results are *Bacillus cereus* (Bce-2) [37] and formulations of *Trichoderma asperellum* (Blite free^®^ CL) and *T. koningiopsis* (RG-160) [27].

## 3. Fusarium Wilt in Strawberry Caused by *Fusarium oxysporum* f. sp. *fragariae*

Another emerging disease affecting strawberry cultivation is Fusarium wilt, caused by *Fusarium oxysporum*. This species is morphologically defined and phylogenetically complex, with its sexual reproduction still unknown. It is a ubiquitous soil fungus, with isolates showing differential pathogenicity towards important crops, although there are also non-pathogenic strains. Pathogenic isolates of *F. oxysporum* affect one or several specific plant species, grouping them into *formae speciales* based on host specificity [38]. The forma specialis associated with strawberries is *F. oxysporum* f. sp. *fragariae* (*Fof*), first identified and described in Australia in 1962 [39], later found in Japan in 1969 [40] and South Korea in 1974 [41]. Reports of this disease increased worldwide in the early 2000s, including in major fruit-producing regions such as California (2006) and Spain (2014) [42,43]. Many of these recent occurrences are linked to changes and restrictions in soil fumigation [44], long-distance pathogen spread via asymptomatic planting material [45], and the species’ ability to acquire virulence through independent evolutionary processes and horizontal chromosome transfer [46,47].

### 3.1. Syndromes

The disease’s symptoms begin with the wilting of older leaves while the younger leaves remain green, accompanied by stunted growth; the plant may eventually collapse entirely (Figure 8a). Cutting the crown usually reveals a brown to black discoloration of the vascular tissues, with more or less extensive necrotic patches in the pith (Figure 8b). 

These symptoms can be accompanied by foliar chlorosis (Figure 9a), starting with the younger leaves (Figure 8b), distinguishing the “yellows-fragariae” pathotypes, which produce chlorosis, from the “wilt-fragariae” pathotypes, which do not. However, the two described syndromes are not specific enough for a definitive diagnosis. 

### 3.2. Pathogenic Diversity

Isolates of the “yellows-fragariae” pathotype carry an accessory chromosome linked to their pathogenicity, with evidence of horizontal transfer. Conversely, this chromosome is not present in “wilt-fragariae” isolates, indicating independent acquisition of pathogenicity. Both pathotypes contain several monophyletic groups [46].

Two races have been described based on the differentiator cultivars ‘Sweet Ann’ and ‘Ventana’, the latter containing the dominant resistance gene to Fusarium wilt, *FW1* [47]. Isolates that break *FW1* resistance include both “yellows-fragariae” and “wilt-fragariae” pathotypes. Isolates causing disease in resistant cultivars (containing the *FW1* gene) are classified as race 2, while isolates causing disease in ‘Sweet Ann’ but not in ‘Ventana’ are classified as race 1 [46]. Other cultivars containing the *FW1* gene include ‘Fronteras’, ‘Portola’, ‘San Andreas’, and ‘UC Eclipse’. More recently, several cultivars and advanced selections resistant to these races have been described, with various genes associated with gene-for-gene resistance [48]. In California, race 1 is widespread, and race 2 has recently been reported on the ‘Portola’ cultivar (containing the *FW1* gene) in fields in Oxnard, California [49].

Diagnosing Fusarium wilt caused by *Fof* in strawberries requires more than merely isolating *F. oxysporum* from symptomatic plants. The symptoms, as previously discussed, are not sufficiently specific. Even if the fungus is isolated from petioles (Figure 10a) and identified morphologically or genetically as *F. oxysporum* (Figure 10b), a definitive diagnosis requires confirming its pathogenicity specific to strawberries. This step is essential because *F. oxysporum* is a soil-inhabiting species that is pathogenic in some cases but more often saprophytic. Therefore, establishing the pathogenicity of the isolate is crucial for accurate diagnosis, either through pathogenicity testing or, in the case of race 1 isolates, using the qPCR probe described by Burkhardt et al. [50].

In Spain during the 2014/15 and 2015/16 seasons, 19 foci of plants with Fusarium wilt, mostly of the cultivar ‘Splendor’, were detected in farms in the municipalities of Moguer, Cartaya, Lepe, Almonte, Villablanca, and San Bartolomé de la Torre [40]. For the isolates obtained, it was found [46] that (i) they belonged to the same vegetative compatibility group; (ii) they were of the “wilt-fragariae” pathotype, less aggressive than the “yellows-fragariae”; (iii) they composed a distinct monophyletic group distant from the groups of the Californian, Korean, Australian, and Japanese isolates; (iv) they were not assigned to either of the two defined races because they did not cause disease in either ‘Sweet Ann’ or ‘Ventana’ but did cause disease in ‘Splendor’, and the genetics of resistance to these isolates in ‘Sweet Ann’ is unknown; and (v) disease occurred in only 3 of the 25 cultivars tested, suggesting that resistance to this group is widespread in strawberry cultivars. These data leave no doubt about the independent origin of *F. oxysporum* f. sp. *fragariae* in Spain and its lower virulence and aggressiveness compared to the isolates present in California.

In recent seasons, technical experts and farmers perceive that the disease’s prevalence and incidence have been low. This is likely due to the low susceptibility of the widely used cultivars in Huelva to the Spanish isolates and the possible attribution of their symptoms to other diseases such as Verticillium wilt or charcoal rot caused by *Macrophomina phaseolina*, particularly at the end of the season. The lack of a rapid molecular identification procedure for this genotype has further complicated the issue for diagnostic laboratories.

However, during the previous season (2023/24), a breeding farm in Isla Cristina (Huelva) was identified with a high incidence of Fusarium wilt, initially attributed to the Spanish genotype. Disease incidence on this farm was widespread, close to 100%, affecting two new advanced selections. Both advanced selections had the cultivar ‘Splendor’ as a parent (Figure 11). This situation highlights the potential risk of this *Fof* genotype to susceptible cultivars.

### 3.3. Sources of Inoculum

Another anticipated issue is the introduction of isolates of the “yellows-fragariae” pathotype from California, due to the transit of plant material by certain plant production companies. During the 2022/23 season, we identified a farm in the Portuguese Alentejo with a high incidence of race 1 “yellows-fragariae” pathotype in a spring planting of the day-neutral cultivar ‘Monterey’ (Figure 12) [51]. The disease first emerged in small patches two seasons before. Given that these fields were previously uncultivated, it is likely that the pathogen was introduced through planting material from Spanish nurseries.

In addition to this finding, our laboratory conducted 488 qPCR analyses for *Fof* race 1 over the last two seasons, detecting 69 positive cases from various Spanish nurseries, which represents approximately 14% of the samples analyzed. These results further underscore the concerning trend of race 1 isolates of *Fof* spreading through our nurseries and fruit production fields (Figure 1). This problem is particularly alarming in light of two recent systematic surveys conducted in different areas of California (Watsonville and Salinas and Santa Maria), which found that 31.1% and 16% of the total soilborne fungal symptom samples corresponded to race 1 of *Fof*, respectively [52,53].

## 4. Conclusions

The cultivation of strawberries in Spain and other regions has been significantly affected by emerging fungal diseases, particularly those that include nursery plants among their sources of inoculum. Notably, in the last few years, *Neopestalotiopsis* spp. and *Fof* have emerged, presenting new challenges to strawberry growers. Several factors had contributed to this emergence, including the prohibition of key soil fumigants, the increased introduction of new cultivars that may be more susceptible, and the international movement of plant material. These diseases, which cause root and crown rot, leaf spot, and, in some cases, complete plant collapse, are also becoming widespread across other major strawberry-producing regions. The complexity of managing these diseases lies in the variability of pathogen aggressiveness, limited diagnostic tools, and the scarcity of effective, authorized chemical treatments. This situation underscores the need for ongoing monitoring and an integrated approach to disease management that combines resistant cultivars, new and effective authorized chemistries and biological controls, and improved diagnostic techniques.

## 5. Future Directions

Future research and management efforts should focus on several key areas:(v)Developing more precise molecular diagnostic tools for identifying the different *Fof* genotypes and different species of *Neopestalotiopsis* spp. will be crucial for early detection and control.(vi)Efforts should continue to develop and introduce strawberry cultivars with stronger resistance to emerging pathogens.(vii)As chemical treatments become more restricted, biological control agents such as *Trichoderma* spp. and *Bacillus* spp. should be explored further as alternatives. In addition, it is essential to integrate cultural practices that reduce the spread of pathogens, such as crop rotation and the use of healthy nursery plants.(viii)Given the global nature of the strawberry trade, international efforts to monitor and control the spread of pathogens are critical. Sharing data on pathogen emergence, resistance patterns, and best practices for disease management across borders will help mitigate risks for growers worldwide.(ix)The impact of changing planting practices and climate, especially higher temperatures, should be further studied. Addressing how these factors contribute to disease susceptibility can guide adjustments in cultivation techniques to reduce stress on plants and lower the incidence of disease.

By addressing these areas, the strawberry industry can develop more effective strategies to combat these and other new emerging diseases and ensure sustainable production.

## Figures and Tables

**Figure 1 plants-13-03441-f001:**
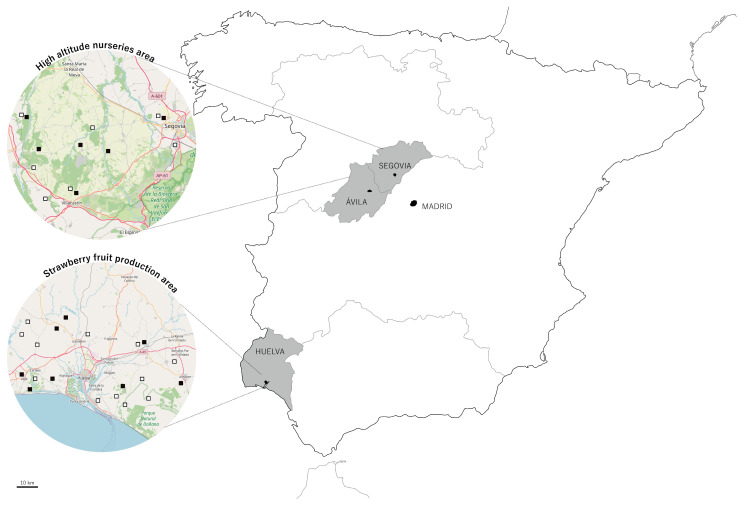
Map of Spain showing nurseries or fruit production fields with plants infected with *Neopestalotiopsis* spp. (▫) and *Fusarium oxisporun* f. sp. *fragariae* race 1(▪) over the last two seasons.

**Figure 2 plants-13-03441-f002:**
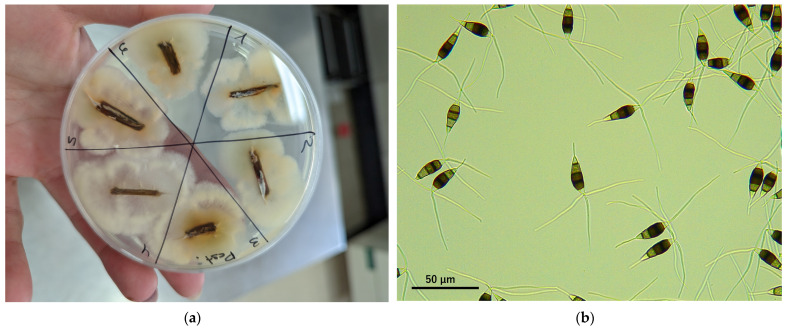
(**a**) Root incubation on microbiological medium with fungal growth of *Neopestalotiopsis* spp.; (**b**) conidia of *Neopestalotiopsis* spp. obtained from conidiomata formed on the incubation medium.

**Figure 3 plants-13-03441-f003:**
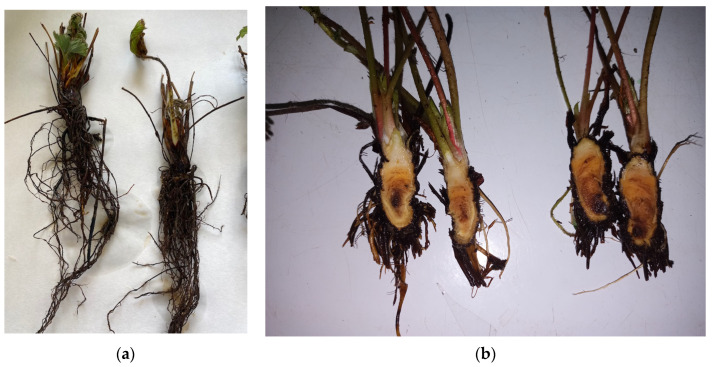
(**a**) Strawberry plants with root rot caused by *Neopestalotiopsis* spp.; (**b**) Strawberry plants with symptoms on crowns caused by *Neopestalotiopsis* spp.

**Figure 4 plants-13-03441-f004:**
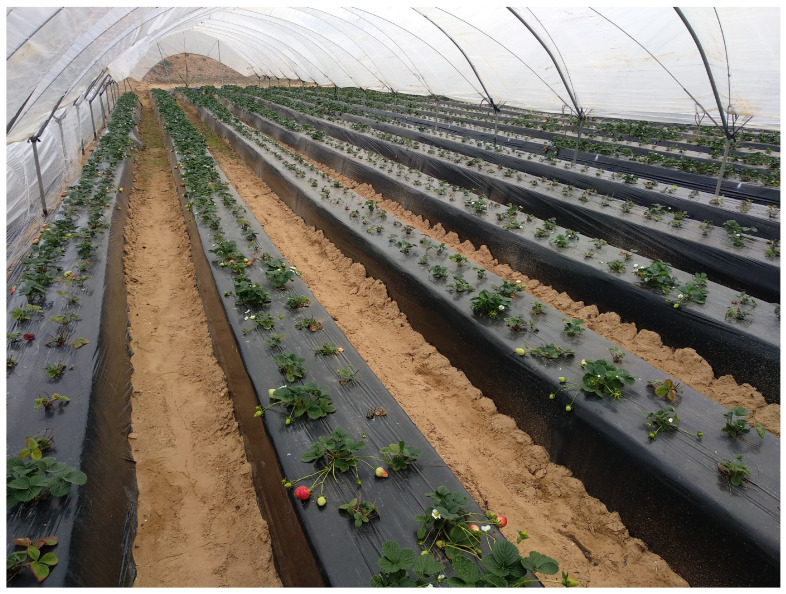
Focus of diseased plants by *Neopestalotiopsis* spp. after transplanting.

**Figure 5 plants-13-03441-f005:**
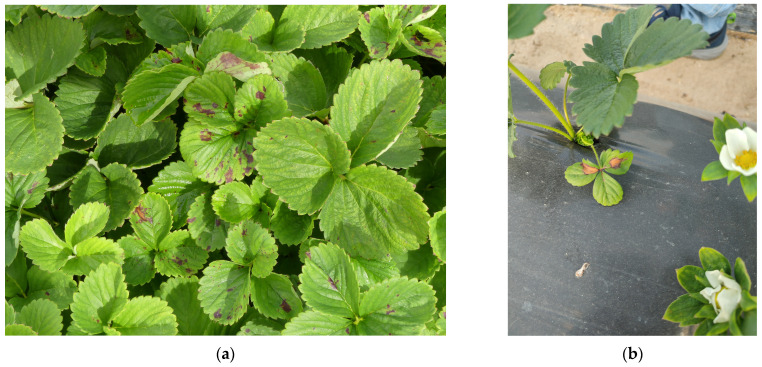
(**a**) Leaves with small lesions caused by *Neopestalotiopsis* spp.; (**b**) leaves with big lesions caused by *Neopestalotiopsis* spp.

**Figure 6 plants-13-03441-f006:**
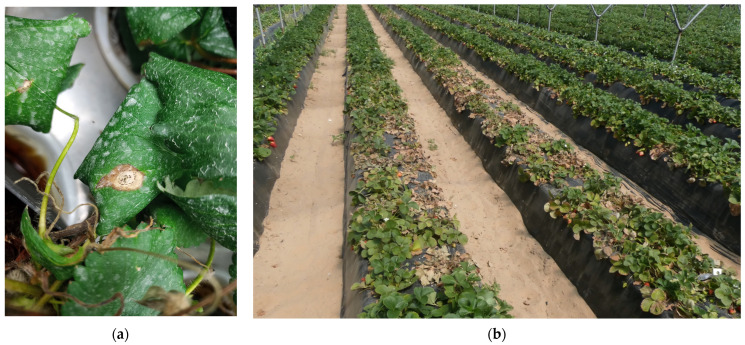
(**a**) Leaf spot with conidiomata of *Neopestalotiopsis* spp. present; (**b**) strawberry beds showing symptomatic plants affected by *Neopestalotiopsis* spp.

**Figure 7 plants-13-03441-f007:**
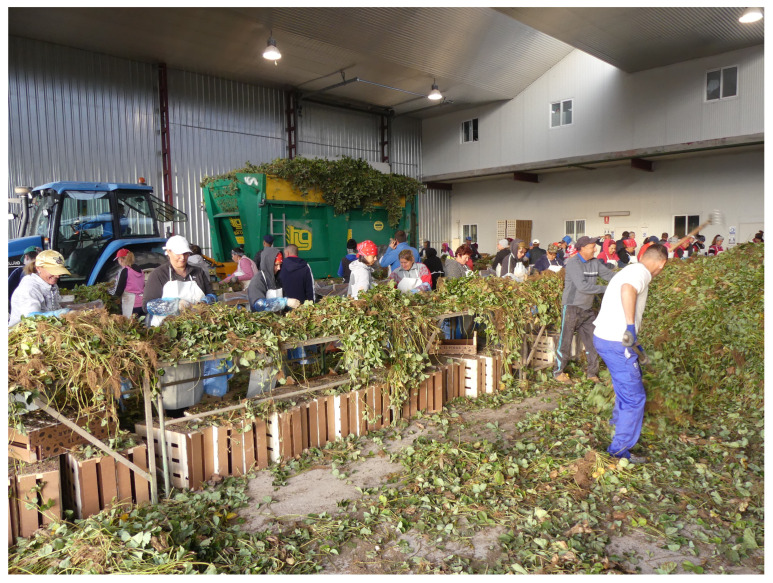
Cutting and handling of the plants in the nursery.

**Figure 8 plants-13-03441-f008:**
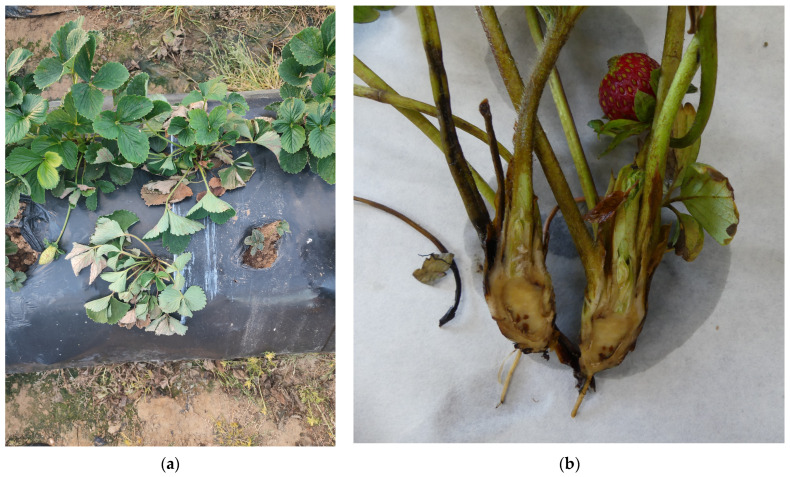
(**a**) Strawberry plants with Fusarium wilt symptoms (pathotype “wilt-fragariae”); (**b**) crown section of a plant with Fusarium wilt showing the necrotic vascular tissue.

**Figure 9 plants-13-03441-f009:**
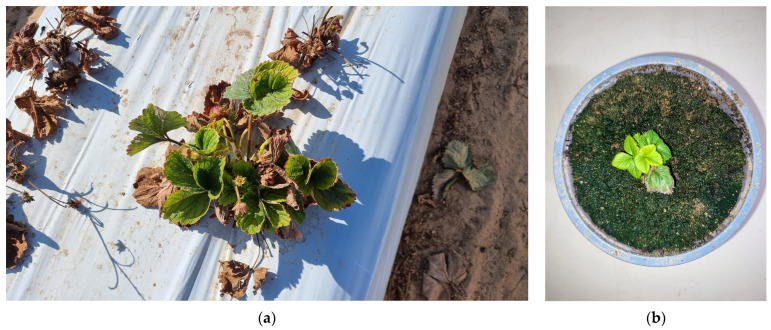
(**a**) Strawberry plants with Fusarium wilt symptoms (pathotype “yellows-fragariae”); (**b**) chlorotic young leaves on a strawberry plant with Fusarium wilt (pathotype “yellows-fragariae”).

**Figure 10 plants-13-03441-f010:**
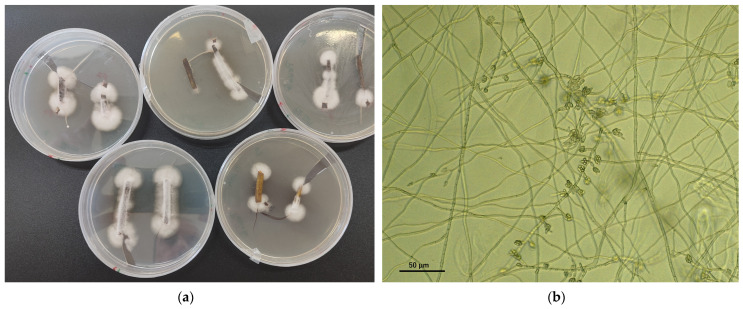
(**a**) Incubation of petioles on semi-selective Komada microbiological medium with fungal growth of *Fusarium oxysporum*; (**b**) microphialides and conidia on false heads of *Fusarium oxysporum* observed in fungal growth.

**Figure 11 plants-13-03441-f011:**
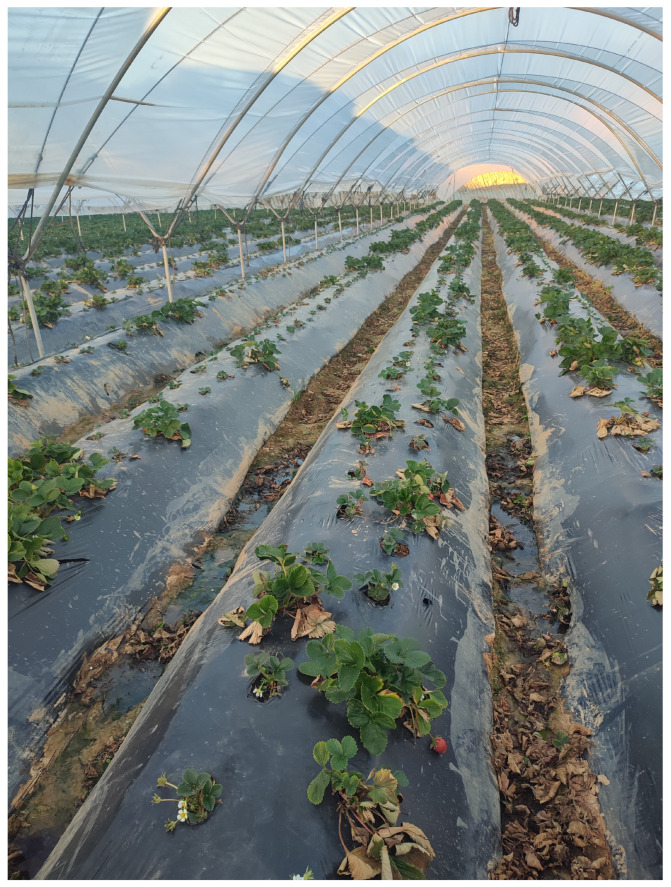
Strawberry plants showing disease symptoms caused by Fusarium wilt (pathotype “wilt-fragariae”).

**Figure 12 plants-13-03441-f012:**
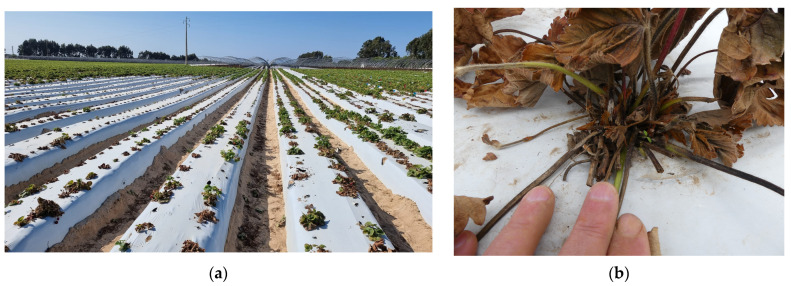
(**a**) Strawberry plants showing Fusarium wilt disease symptoms (pathotype “yellows-fragariae”); (**b**) strawberry plant with symptoms of Fusarium wilt showing sporodochia at the base of petioles (pathotype “yellows-fragariae”).

## Data Availability

The data presented in this study are available on request from the corresponding author due to legal reasons.

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
