# Peer review of "Emerging Diseases in Spain Strawberry Crops: Neopestalotiopsis Leaf and Crown Rot and Fusarium Wilt"

_plants, 2024, doi:10.3390/plants13233441_

Round 1

Reviewer 1 Report

Comments and Suggestions for Authors

The manuscript is well-structured and written, with a clear introduction, relevant figures, a concise conclusion, and potential future research. 

The manuscript provides a good overview of strawberry diseases prevalent in Spain, discussing their origins, characteristics, and potential impacts on agriculture. It well cites the existing literature to offer insights into the current understanding of these diseases. However, I would have expected to see a short section on the economic impact of these diseases.

Based on the analysis and perspectives provided, it is a good fit for publication as a perspective article. It offers valuable contributions to the understanding of strawberry diseases in Spain and will be of interest to researchers and practitioners in the field.

Reviewer 2 Report

Comments and Suggestions for Authors

This article provides important information about the emerging fungal diseases affecting strawberry cultivation in Spain. However, the text could benefit from improvements in clarity, coherence, and academic tone. The authors must address the following questions before the manuscript can be considered for publication.

The manuscript lacks a description of the disease occurrence regularities and the pathogen. A detailed description of the pathogen's life habits and infection patterns should be provided. This will aid in the precise prevention and control of the disease.

The figure quality is too poor; the unnecessary parts should be cropped.

The manuscript should cite more high-quality papers to ensure the reliability of the paper.

Abstract: The core areas and incidence rates of major emerging diseases in Spain should be specified.

Introduction: The dangers brought by these diseases should be detailed, as well as the objectives and significance of this paper. Additionally, the introduction section should have sufficient literature support.

Lines 88 and 175: The introduction of these two sections lacks clarity in terms of logic. It should be introduced under secondary headings in sequence, such as: 2.1 Symptoms, 2.2 Disease Distribution and Incidence Rate, 2.3 Prevention and Treatment. Additionally, in Figure 1 and Figure 9, do the colonies have spore-forming structures and spores? If so, photos of these structures should be included.

Reviewer 3 Report

Comments and Suggestions for Authors

In this manuscript, the diseases caused by Neopestalotiopsis spp. and Fusarium oxysporum f. sp. fragariae in Spain mainly are introduced. So the title of the manuscript is too big. The title should be in more detail.

The focused diseases occur in Spain, so the geographic distribution of the diseases should be provided in the manuscript, especially, maps are the most needed. Most words in this manuscript are described qualitatively. The data on incidence, severity and risk of the diseases should be provided in the manuscript. The photos of the pathogens should be provided. There are too many keywords, so some keywords should be deleted. In line 19, ‘f. sp.’ should be not in italics. There is difference between ‘Castilla y León’ in line 31 and ‘Castilla and León’ in line 33? Delete ‘)’ in line 32. The words in lines 116-118 should be merged into the previous paragraph?

Round 2

Reviewer 2 Report

Comments and Suggestions for Authors

  This is better but there is still room for improvement. I believe the author has not made all the changes we pointed out in the updated manuscript yet. The issues regarding the revision of the abstract and the clarification of the logic in the main text, which we raised, have not been fully addressed. There are significant issues with the figures. First and foremost, a scale bar should be added to the figures of the conidiophores, and the conidiophores in Figure 10 are not clear enough. Figure 2, 3, 5, 6, 8, 9, 10, and 12 should each be arranged without any gaps between them, and 'a' and 'b' should be uniformly labeled on the figures.

Reviewer 3 Report

Comments and Suggestions for Authors

1. The focused diseases occur in Spain, so the geographic distribution of the diseases should be provided in the manuscript, especially, maps are the most needed. Why only the host plant producing areas were provided in Figure 1? Moreover, the map in Figure 1 is not a standard map, without the scale and other map elements. These should be revised. 2. The scale bars should be added to the provided photos of the pathogens.
